# Pharmacodynamic Effect of Luteolin Micelles on Alleviating Cerebral Ischemia Reperfusion Injury

**DOI:** 10.3390/pharmaceutics10040248

**Published:** 2018-11-29

**Authors:** Liwei Tan, Chen Liang, Yeye Wang, Yu Jiang, Shengqiao Zeng, Rui Tan

**Affiliations:** 1College of Medicine, Southwest Jiaotong University, Chengdu 610031, China; rafael0927@163.com (L.T.); 18780163811@163.com (C.L.); yezi_1317@163.com (Y.W.); jiangyu950704@163.com (Y.J.); zzshqiao@gmail.com (S.Z.); 2College of Life Science and Engineering, Southwest Jiaotong University, Chengdu 610031, China

**Keywords:** nanomedicines, luteolin-loaded micelles, cerebral ischemia reperfusion injury, bioavailability, antioxidative stress

## Abstract

Oxidative stress and inflammation are important mechanisms of cerebral ischemia reperfusion (IR) injury. Luteolin (Lu), one of the major active components in the classical Tibetan prescription, which has been used in the treatment of cardiovascular diseases since 700 BC, has potential for IR injury therapy. Its hydrophobicity has impeded its further applications. In this study, we first prepared Lu micelles (M-Lu) by self-assembling with an amphiphilic copolymer via the thin film hydration method to improve the dispersion of Lu in water. The obtained M-Lu was about 30 nm, with a narrow particle size distribution, and a 5% (*w*/*w*) of Lu. The bioavailability of the micelles was further evaluated in vitro and in vivo. Compared to free Lu, M-Lu had a better penetration efficiency, which enhanced its therapeutic effect in IR injury restoration. M-Lu further strengthened the protection of nerve cells through the nuclear factor-κ-gene binding κ (NF-κB) and mitogen-activated protein kinases (MAPK) pathways and inhibited the apoptosis of cells by adjusting the expression of B-cell lymphoma-2 (Bcl-2) and Bcl-2 associated X protein (Bax) in the case of oxidative stress damage. M-Lu induced stem cells to differentiate into neuron-like cells to promote the repair and regeneration of neurons. The results of in vivo pharmacodynamics of Lu on occlusion of the middle cerebral artery model further demonstrated that M-Lu better inhibited inflammation and the oxidative stress response by the down-regulation of the inflammatory cytokine, including tumor necrosis factor (TNF)-α, interleukin (IL)-1β, and IL-6, and the up-regulation of the activity of anti-oxidant kinase, such as superoxide dismutase (SOD) and glutathione peroxidase (GSH-px), which further ameliorated the degree of IR injury. The M-Lu could be a new strategy for IR injury therapy.

## 1. Introduction

Cerebral ischemia reperfusion (IR) injury generally causes permanent deterioration of the central nervous system (CNS), which is a main cause of death and the most common neurological disease resulting in long-term disability worldwide [1,2,3]. 

Until now, a large number of studies indicated that some pathological features are associated with IR injury, including the production of radical oxygen species, calcium overload, energy failure, cell apoptosis, and inflammatory reaction. They precipitate the reduction in neuronal death and neurological dysfunction [4,5,6]. Among these pathological mediators, inflammation and oxidative stress play important roles in the pathogenesis of IR injury [7,8]. In general, inflammation is a complex biochemical reaction to maintain homeostasis to the harmful stimuli [9]. However, the upregulation of pro-inflammatory cytokines and inflammatory mediators, such as nitric oxide, TNF-α, and IL-1, among others, poses similar cytotoxic threats to the invading pathogens and the host cells. Prolonged inflammation could also lead to serious tissue injury [10]. Accompanying the inflammatory response, oxidative stress is another accelerator in the pathogenesis of IR injury. Growing experimental evidence indicates that the failure of metabolic reactions during IR leads to the elevation of radical oxygen species (ROS), which further induces oxidative damage [11,12]. As a result, inflammatory response and oxidative stress commonly promote apoptosis of the nerve cells and permanently damage the CNS [13]. Some reports indicated that endogenous stem cells and progenitor cells could differentiate into neuron-like cells via interactions with the micro-environment to promote the neurological function recovery, which is the other important mechanism in neuroprotection [14]. Therefore, therapeutic strategies targeting the multiple pathogenic factors would be feasible methods for managing such a complicated pathological injury.

Some traditional formulations with multiple components produced significant therapeutic outcomes on the clinical application of IR injury [15]. Thus, developing effective anti-IR drugs by evaluating the pharmacodynamics mechanism of the active components in traditional formulations may be of value [16]. In our previous study, a classical traditional Tibetan formulation, called the “Ruyi Zhenbao” pill, which has been applied in the cerebrovascular diseases treatment since 700 BC, was comparatively analyzed for the presence of common active components in plasma after oral administration. A natural active molecule—luteolin (Lu)—was recognized in this complex formulation, as shown in Figure 1. The results revealed that Lu might be one of the main active components of this traditional formulation.

Luteolin is abundant in fruit, vegetables, and medicinal herbs. It is an anti-oxidant, which also has anti-inflammatory, neuroprotective, and anti-allergic properties. It alleviates the progression of neurodegenerative and some potential damaging diseases [17,18]. However, the side effects caused by hydrophobicity include low-selective distribution, low bioavailability, and fast elimination, which have restricted the wider application of Lu [19,20]. Therefore, in order to improve the therapeutic effect and systematically evaluate the mechanisms of Lu in IR injury treatment, constructing a stable drug delivery system (DDS) without physical-chemical characteristics changing the drug molecule is necessary.

With the development of nanotechnology, nano-formulations provide an alternative strategy for IR injury therapy [21]. Compared to the traditional formulation, nano-formulation not only ameliorates the hydrophilicity, prolongs the circulation time, and realizes the controlled release properties of drugs, but can also actively target drug delivery sites by special surface modification [22]. Up to date, more than 40 kinds of nano-formulations have been approved for study in clinical trials [23]. However, most of the approved nano-formulations were applied in cancer therapy. So, nano-formulations for IR treatments require research and development. In various nano carriers, the amphiphilic copolymer poly(ethylene oxide) monomethyl ether-poly(lactide-*co*-glycolide) (MPEG-PLGA) has been approved by Food and Drug Administration (FDA) as pharmaceutical excipients, attributed to its superior drug loading capacity and biological safety [24]. Since the permeability of the blood-brain barrier (BBB) is open during the acute phase of IR injury, which provides a channel for the drug to penetrate into the focal area, it is critical to overcome the disadvantages of Lu to enhance its enrichment in the injured area [25]. Therefore, MPEG-PLGA micelles are an ideal carrier for Lu. The core of the micelle consists of the hydrophobic block PLGA, which prevents Lu from being recognized by the album serum protein, and the MPEG shell reduces the eliminating of drugs by reticuloendothelial system (RES) [26].

Therefore, in this study, we first prepared Lu micelles (M-Lu) by the film hydration method, as described in Scheme 1. Then, the properties of Lu-M, including drug loading (DL), encapsulation efficiency (EE), size, morphology, and the in vitro release profile, were further characterized. The pharmacodynamic effects of M-Lu were investigated in vitro and in vivo. Considering that Lu-M in this study was mainly used for IR injury treatment, the relative indicators, such as inflammatory mediator, production of oxidative stress, apoptotic mediators, and neuroprotective effect, were investigated in detail. Finally, the therapeutic efficacy of the M-Lu in ameliorating IR injury were evaluated on the occlusion of middle cerebral artery (MCAO) model. All results demonstrated that Lu could ameliorate IR injury, and the nano-formulation could improve the therapeutic effect and be a candidate for the IR injury treatment.

## 2. Materials and Methods

### 2.1. Materials

Ruyi Zhenbao pills were supplied from Balaqushenshui Tibetan Pharmaceutical Company (Tibet, China), and the luteolin was supplied from Pusi Biotechnology Co. Ltd. (Chengdu, China). The block copolymer MPEG-PLGA (Da: 2000-2000) was provided by Xi’an Ruixi Biological Technology Co., Ltd (Xi’an, China). Dulbecco’s modified Eagle’s medium (Sigma-Aldrich, Saint Louis, MO, USA) (DMEM) and 3-(4,5-dimethylthiazol-2-yl)-2,5-diphenyltetrazolium bromide (Sigma-Aldrich) (MTT) were used without any further purification. DAPI Staining Solution (DAPI), Coumarin-6 (C6), and all the high performance liquid chromatography (HPLC)-grade reagents, acetonitrile, methyl alcohol, and ethyl alcohol were purchased from Sigma-Aldrich. 2,3,5-triphenyltetrazolium chloride (TTC) was obtained from Sigma-Aldrich. For Western blot assay, antibodies against B cell leukemia-2 (Bcl-2), Bcl-2-associated X protein (Bax), nuclear factor-k-gene binding κ (NF-κB,) phosphor-p38 (p-p38), phosphor-ERK1/2 (p-ERK1/2), and phosphor-c-Jun N-terminal kinase (Thr183/Tyr185) (p-JNK), TNF-α, IL-6, IL-1β, the β-actin, neuron-specific enolase (NSE), Nestin, and Glial fibrillary acidic protein (GAFP) antibody, and horseradish peroxidase-conjugated secondary antibodies were obtained from Servicebio Technology Co., Ltd (Wuhan, China). 

Pheochromocytoma cell (PC12 cell) line was obtained from the American Type Culture Collection (ATCC; Rockville, MD, USA), and grown in DMEM. The cell culture was maintained in a 37 °C incubator with a humidified 5% CO_2_ atmosphere. 

Male Sprague-Dawley (SD) rats were purchased from Dashuo Bio-Technology. Co. Ltd. (Chengdu, China). The rats were housed at a temperature of 20 ± 2 °C, relative humidity of 50–60%, and with 12 h light-dark cycles. Animal experiment procedures were performed according to the protocol approved by the Institutional Animal Care and Use Committee of Sichuan University (IACUC-S200904-P001).

### 2.2. Active Compounds in Ruyi Zhenbao Pill for Analysis

The active compounds in Ruyi Zhenbao pills were detected in vivo by liquid chromatograph-mass spectrometer (LC-MS) and HPLC (HPLC 1260, Agilent, Santa Clara, CA, USA) with C18 column (4.6 mm × 250 mm × 5 μm, Ultimate^®^ XB-C_18_). The in vivo analysis of compounds was obtained as follows: the rats were fasted overnight prior to drug administration and randomly divided into three groups. The rats were treated with Ruyi Zhenbao pill in normal saline orally by gastric intubation. Two hours later, blood samples were collected and immediately centrifuged to obtain the plasma. The active compounds were extracted from plasma by methanol. The organic layer was evaporated and reconstituted in mobile phase analysis.

### 2.3. Preparation and Characterization of M-Lu

M-Lu was prepared by the thin-film hydration method [26]. Firstly, Lu and MPEG-PLGA were co-dissolved in 4 mL of dehydrated alcohol, and then the solvent was evaporated in rotator evaporator at 60 °C to obtain the homogenous film. Next, the mixture was hydrated in double distilled water at 60 °C to self-assemble into micelles. Finally, the micelles were filtered by a 0.22 μm syringe filter (Millex-LG, Millipore Co., Boston, MA, USA) and freeze-dried to produce M-Lu powder. The particle size distribution was determined by dynamic light scattering (DLS) (Malvern Nano-ZS 90, Malvern, UK), and the morphology was observed by transmission electron microscopy (TEM). Due to the hydrophobility of Lu, the dimethyl sulfoxide (DMSO) solution of Lu (free Lu) was used as the reference preparation in this study. 

### 2.4. DL and EE of M-Lu 

The DL and EE of the micelles were determined by HPLC with a C18 column (4.6 mm × 250 mm × 5 μm, Grace Analusis column). The compositions of the mobile phase were phosphoric acid solution (0.2%)/methanol solution (45/55, *v*/*v*) at a flow rate of 1 mL/min. Detection was recorded on a diode array detector (1260 DAD VL) at a wavelength of methanol solution (nm). Before measurement, the samples were diluted with 75% methanol solution. The results were calculated using the following equations:DL% = Amount of Lu determined in micelle/(Amount of Lu determined + copolymer) × 100%
EE% = Amount of DTX determined in micelle/Amount of Lu in feed × 100%

#### 2.4.1. In Vitro Drug Release Study 

The release behaviors of the M-Lu were investigated using the dialysis method. Free Lu and M-Lu were separately suspended in a dialysis bag (molecular weight cutoff: 8000 Da), and were then immersed in phosphate buffered saline (PBS) solution with 0.05% (*v*/*v*) Tween 80 at 37 °C under horizontal shaking (100 rpm/min). At predetermined intervals, the aliquots (2 mL) were withdrawn and replaced by the same amount of PBS. The amount of Lu released at designated time points were measured by HPLC. The chromatographic conditions and the methods followed determination of DL.

#### 2.4.2. Cell Uptake Study

A hydrophobic fluorescent C6 probe was used instead of Lu to explore the cellular uptake of Lu. The PC12 cells were incubated in 6-well plates for 24 h at 37 °C, and the free C6 and M-C6 (concentration of C6 was 100 ng/mL) were added to the plates. After 4 h incubation, the cells were washed by PBS three times and fixed with 80% alcohol. Finally, the fixed cells were treated with DAPI for 5 min. The fluorescence images of cellular uptake were observed under a laser confocal fluorescence microscopy (LCFM) (A1si+/A1Rsi+, Nikon, Tokyo, Japan). 

### 2.5. Cell Viability

The cell viability of MPEG-PLGA, free Lu, and M-Lu were evaluated by MTT assay on the PC12 cell line. In these tests, the cells were seeded in 96-well plates at a density was 1 × 10^4^ cells per well for 24 h incubation. The free Lu, M-Lu, and MPEG-PLGA at different concentrations were added into the plates for 4 h co-incubation. Then, H_2_O_2_ was added into the free group and M-Lu group for another 24 h incubation. Next, 20 μL MTT solution (5 mg/mL) was added to each well, and the solution was replaced with DMSO (160 μL per well) after 3 h. The absorbance was measured by a 680-model microplate reader from an infinite M200 microplate reader (Tecan, Durham, NC, USA).

### 2.6. Western Blot

The mechanism of improving the cell injury was investigated by Western blot (WB). Firstly, the PC12 cells in different groups were homogenized at −80 °C. Next, the samples were lysed in Radio-Immunoprecipitation Assay (RIPA), the supernatant and equal amounts of protein were loaded on dodecyl sulfate, sodium salt (SDS)-Polyacrylamide gel electrophoresis (SDS-PAGE) gel. The gels were transferred to poly (vinyl formal) (PVDF) membranes during night after SDS-PAGE, and then blocked with 5% milk in tris buffered saline (TBS). The rat monoclonal antibody to NF-κB, p-JNK, p-P32, p-ERK1/2, Bcl-2, and Bax was used as the primary antibody, and the goat anti-rat horseradish peroxidase (HRP)-labeled antibody was used as the secondary antibody. The protein expression on membranes was analysis with an exposure meter.

### 2.7. Neuroprotective Effect of Lu In Vitro

The bone mesenchymal stem cells (bMSC) from tibial bone marrow cavity of SD rats (3–4 weeks old) were used to evaluate the neuroprotective effect of Lu in vitro. The cells were seeded into 6-well plates for 24 h incubation. Then, free Lu and M-Lu were added into the plate for another 24 h incubation. After the preprocessing, including fixing and blocking, the rat monoclonal antibody to Nestin, GFAP, and neuron-specific enolase NSE, used as the primary antibody, were incubated. Finally, the cells were treated with appropriate secondary antibody to achieve the immunofluorescence, which could be detected by the LCFM.

### 2.8. Construction of MCAO Model 

As performed previously, rats were first fixed in a supine position after being anesthetized by choral hydrate (10%, 100 g/mL, intraperitoneal injection (i.p.)). Secondly, one side of the jugular vein and bilateral carotid arteries were exposed. The reperfusion of the blood was followed the removing of artery clamps after 2 h occlusion of the middle cerebral artery. Finally, the degree of nerve damage was evaluated according to the Zea Longa. Except for the middle cerebral artery occlusion and hemostasis from the common vena jugulars, the same operation above was performed on the rats in the sham group. Animal experiment procedures approved by the Institutional Animal Care and Use Committee of Sichuan University in in September 2014. All animal experiments were carried out in compliance with guidelines.

### 2.9. Measurement of Cerebral Infarct Volume and Water Content

Rats were sacrificed at 24 h after administration, and their brains were collected for measuring infarct area (*n* = 5) and water content. The brains were removed immediately and washed with phosphate buffer solution (PBS, 0.1 mol/L) three times. Then, a 2 mm slice of the brain tissue was immersed in 2% TTC (*v*/*v*) stain for 15 min. The white and red area represent the infract area and non-infract area, respectively. The infarct volume was calculated as follows:The infarct volume (%) = Weight of white area/Weight of whole brain × 100%

For the measurement of water content, the weights of fresh tissues (*W*_1_) were measured with an electronic balance. After 8 h of moisture removal, weights of the samples (*W*_2_) were measured again. The ratio of the water content in each group was calculated using the following formula:Ratio of the water contents (%) = (1 − *W*_2_/*W*_1_) × 100%

### 2.10. BBB Disruption Evaluation

Evans blue (EB) leakage was used to investigate the BBB disruption after IR injury. We injected 2% EB (*v*/*v*) solution in PBS via the tail vein 4 h after reperfusion (4 mL/kg). Then, 24 h after administration, mice were sacrificed, and the images of the EB leakage into the ischemic brain were obtained.

### 2.11. Drug Delivery In Vivo

To investigate the delivery of the micelles in vivo, MCAO model rats were sacrificed 4 h after injection with free coumarin-6 (C6) and M-C6 (C6 micelles) though tail veins. The brain tissues were removed and cut into 7-μm slices for observation under fluorescence microscope (Olympus, Shanghai, China) after embedded in an optimal cutting temperature (OCT) compound. The dose of C6 was 2 mg/kg.

### 2.12. Histopathology Tests

The brain tissues were fixed in 4% paraformaldehyde for 48 h. Then, the samples were embedded in paraffin and coronal sectioned around 7 μm for a series of detection. The thicknesses were stained with hematoxylin and eosin (H&E) stain. The structures of the brain tissues were observed under microscope (Olympus, Shanghai, China). The injured neurons cells were stained dark, shrunken or dysmorphic, with intact distinct nucleus and nucleolus. For Nissl staining, tissues sections were dehydrated in ethanol and chloroform, and then stained with 1% toluidine blue for 10 min. At last, the sample were coverslipped after clearing in graded ethanol and xylene. Additionally, the apoptosis of neurons was evaluated by terminal deoxynucleotidyl transferase (dUTP) nick end-labeling stain (TUNEL) assay according to the manufacturer’s instructions of the TUNEL kits. The counts of cells were performed with a magnification microscope.

### 2.13. Biochemical Analysis

The activity of inflammation factors was evaluated based on the expressed level of TNF-α, interleukin-6 (IL-6), and interleukin 1-β (IL-β). The samples were coronal sectioned around 7 μm for a series of detection. The immunofluorescence antibodies were specially connected to the target protein, and the methods followed the instruction for use. DAPI was used to exhibit the area of the cells. The fluorescence was observed under a fluorescence microscope. The content of malondialdehyde (MAD), GSH-px, and the activities of SOD were investigated for evaluating the anti-oxidative stress effect of Lu. The samples were prepared follow the instructions in the MAD, GSH-px, and SOD kits, and the contents were measured with an automatic biochemical analyzer (Rayto life and analytical sciences Co., Ltd, Shenzhen, China).

### 2.14. Statistical Analysis

The comparison of each group was evaluated by the statistical analysis using SPSS software with one-way ANOVA. All the statistical results are expressed as the mean ± SD, and a *p*-value < 0.05 was considered statistically significant.

## 3. Results and Discussion

### 3.1. Preparation and Characterization of M-Lu

IR injury is an important cause of death and long-term disability. Based on the many reports and our previous study, Lu is a natural flavone with neuroprotective effects, which is one of the main active components in the traditional formulation that has been used for the treatment of cardiovascular diseases including IR injury. Therefore, Lu would be a candidate for managing IR injury after some modification to improve its disadvantages, such as hydrophobicity. 

To improve the Lu solution, we entrapped the molecule into the amphiphilic copolymers MPEG-PLGA using the thin film hydration method [27]. MPEG-PLGA and Lu were distributed as homogenous amorphous substances after the evaporation process was used to remove the dehydrated alcohol. Then, the copolymers could be self-assembled into core-shell structured micelles with Lu encapsulated in the core after hot water was added at 60 °C. Finally, the micelles were filtered through a 0.22-μm syringe filter before being freeze-dried. As shown in Figure 2A, the M-Lu presented as a slightly yellow solution. The mean particle size was about 30 nm and the polydispersity index (PDI) was less than 0.2, which indicated the good dispersity of the micelles. Under TEM (Figure 2B), the morphology of M-Lu was observed as homogeneous spheroid, and the observed size was consistent with the measurements. 

The DL and EE of M-Lu with different drug feeding are displayed in Table 1, and the EE was kept above 95% as drug feeding increased from 2% to 5%. With further increases in drug feeding from 8% to 10%, the EE decreased from 87.62% ± 3.43% to 74.30% ± 2.31%. We selected the M-Lu with a DL about 5%, and the EE of 97.20% ± 2.34% for further investigation. These results suggested that the M-Lu micelle was produced successfully.

### 3.2. Stability and Drug Release Behavior of M-Lu In Vitro

Stability is an important characteristic for determining the therapeutic performance of micelles [28]. In general, the change in particle size is the gold standard for the evaluation of micellar stability. The particle size of M-Lu at 25 °C and 37 °C were determined after re-dissolving. As shown in Figure 2C, the size of M-Lu remained homogeneous in the initial 12 h. At longer storage times, compared to M-Lu at 25 °C, the size and PDI of M-Lu at 37 °C increased sharply. These results suggest that the stability of M-Lu is affected by storage temperature, which might due to degradation of the copolymers being promoted by a relatively high temperature [29]. Additionally, the stability of M-Lu in different media, including normal saline (NS), PBS (pH 7.4), and DMEM, was studied (Figure 2D). The results suggest that that there was no significant difference in particle size between the groups, and the trend was consistent. The in vitro release behavior of the free Lu and M-Lu were investigated for determine the release profile in vivo, and the results are shown in Figure 2E. In the free Lu group, nearly 95% of Lu was released into the release medium within 24 h, and the M-Lu group released 68%. This suggests that micelles could prevent the initial burst release. This property of M-Lu might enable long-term circulation and bioavailability in vivo.

### 3.3. In Vitro Drug Delivery

Increasing the uptake rate of cells effectively improves the therapeutic effect of drugs. To investigate the transport ability of micelles in vitro, Lu was replaced by hydrophobic fluorescent probe C6 to be entrapped into the core of micelles, and the cellular uptake was analyzed under a fluorescence microscope. From Figure 3A, compared to the free C6 group, obvious was observed in the cytoplasm. This suggests that micellization would effectively promote cellular uptake for hydrophobic drugs.

### 3.4. MTT Assay

The cytotoxicity of the carrier was evaluated using the MTT assay on PC 12 cells, and the results are exhibited in Figure 3B. Hypotoxicity of the copolymer for PC 12, even at high concentrations (1000 μg/mL), showed the copolymer is safe. To explore the protective effect of Lu in vitro, we first constructed an oxidative stress damage model in vitro through cultured PC 12 cells with H_2_O_2_. Then, free Lu or M-Lu were added into the cell supernatants and incubated for another 4 h. Finally, the viable ratio of PC 12 cells reflected the protective effect of Lu. From Figure 3C, with increasing Lu concentration to 10 μM, the cell viability reached around 80%, and the M-Lu group exhibited superior activity compared to the free Lu group, which might be contributed to cellular uptake efficiency of the micelles. The cell viability decreased with increasing dosage, suggesting a dose-dependent anti-oxidant stress effect of Lu.

### 3.5. In Vitro Protective Mechanism of Lu 

To clarify the mechanism of the protective effect of Lu on the oxidative stress damage to cells, some relevant cytokines were determined. NF-κB and mitogen-activated protein kinase (MAPK) are two major signaling pathways that caused damage by the expression of some inflammatory cytokines after oxidative stress [30]. MAPK, as one family of serine/threonine kinases, can respond to various stimuli, including oxidative stress, and participate the progression of cell apoptosis [31]. The three distinct groups of MAPKs, including C-Jun N-terminal kinase (JNK), P38 kinase, and extracellular signal-regulated protein kinases 1 and 2 (ERK1/2), play an important role in protecting cells from damage. The states of protein phosphorylation represented the activity of the MAPK single pathway [32,33]. According to Figure 4C, the expression of these three phosphor-proteins in the free Lu group and M-Lu group were significantly inhibited. The expression of p-JNK, p-P38, and p-ERK1/2 in the M-Lu group were lower than in the free Lu group. 

As another signal pathway for inflammation, the activation of NF-κB was measured. Figure 4B shows that M-Lu dramatically down-regulated the expression of NF-κB in oxidative stress injury models. These results suggest that the inflammatory response could be reduced by the inhibition of the activation of MAPK and NF-κB pathways. Apoptosis is a following step of the damage mechanism after IR injury. As regulators of apoptosis, Bcl-2 family proteins, including the pro-apoptotic proteins, Bax, Bad, and Bim, and the anti-apoptotic proteins, Bcl-2 and Bcl-x, regulate the permeability of the mitochondrial membrane to affect intracellular apoptotic signal transduction. The apoptosis of the neurons is related to the ratio of the expression of Bcl-2 and Bax [34]. From Figure 4A, with the addition of free Lu or M-Lu, the expression of Bcl-2 was up-regulated, whereas Bax was down-regulated, illustrating that apoptosis of neurons can be inhibited by Lu through the ratio regulation of the expression of Bcl-2 and Bax in the oxidative stress injury model in vitro. Furthermore, we found that the protective effect of M-Lu was superior to that of free Lu, which might be attributed to the improvement in the cellular uptake for Lu by micellization. 

### 3.6. The Neuroprotective Mechanism of Lu In Vitro

Introducing endogenous stem cells and progenitor cells differentiating into neuron cells to promote the neurological function recovery is an important mechanism in neuroprotection. In this study, bMSC was used to determine the neuroprotective mechanism of Lu in vitro. As shown in Figure 5, the specific neuronal marker proteins, including Nestin [35], GAFP [36], and NSE [37], were observed after the introduction, and the fluorescence in the Lu-M group was brighter than in the free group, which indicates that Lu could introduce bMSC differentiation into neuron-like cells in vitro, and the efficiency could be promoted by micellization. The induction of stem cells by Lu may be due to its ability to up-regulate the expression of the brain-derived neurotrophic factor, which has already been proven to induce stem cells to differentiate into neural cells [38]. These results further illustrate the mechanism of the neuroprotective effect of Lu.

### 3.7. Drug Delivery in MCAO Models

Based on the in vitro results, we further examined M-Lu in vivo for IR injury therapy. MCAO models were established to comprehensively evaluate the therapeutic mechanism of M-Lu. The drugs were administrated two hours after IR injury via tail intravenous injection. Initially, to investigate the delivery efficiency of Lu in vivo, similar to the cellular uptake experiments, C6 was used to replace Lu as a fluorescence probe to be entrapped in the core of the micelle. At four hours after administration, the brains from the MCAO models were collected. Next, the samples were prepared into thin slices about 7 μm thick. As shown in Figure 6B, the green fluorescence of C6 was easily observed, which illustrated that Lu could infiltrate the injured brain tissues. Considerable evidence supported the BBB permeability enhancement in the window of IR injury. This phenomenon was demonstrated in the study once again (Figure 6A). Therefore, the permeability enhancement may promote infiltration efficiency. The intensity of fluorescence in the M-C6 group was much stronger than in the free C6 group. This experimental phenomenon was dependent on the enhancement of the carrier-mediated drug delivery efficiency, which might be caused by surface PEGylation. The PEG molecule on the surface of the micelles could prevent RES recognition in vivo, endowing the micelles with a longer circulation time [26]. Therefore, the hydrophobic micelles-delivered drug could accumulate in damaged brain tissues.

### 3.8. Therapeutic Effect of M-Lu in MCAO Model

We further investigated the in vivo therapeutic effect of M-Lu for anti-IR injury. Forty SD rats were randomly assigned into four groups (10 rats per group), including sham group, IR group (as the negative model, and treated with NS during the experiment), free Lu group, and M-Lu group. The MCAO model rats in the therapeutic groups were treated with a dose of 10 mg/kg free Lu or M-Lu, and the tissues were collected 24 h after administration for further analysis. The results are exhibited in Figure 7. The infarct area measured by TTC stain, and the ratio of infarct volume to the whole brain tissue, were 0% (sham), 28% ± 1.2% (IR group), 21.5% ± 1.83% (free Lu), and 19.2% ± 0.87% (M-Lu group). These results indicate that Lu could reduce the damaged brain volume caused by IR. As shown in Figure 7C, the brain water content in each damaged group did not show a significant difference, which might be cause the BBB permeability could not recover within 24 h. 

### 3.9. Histopathological Study

To verify the mechanism of Lu and M-Lu in vivo, some histopathological studies on the penumbra were carried out, which is the main area affected by IR injury. From the H&E stain results (Figure 8A), we found an unclear and unintegrated tissue structure, different degrees of interstitial edema, and some cellular swelling at the ischemic penumbra in the experimental groups. The brain structure exhibited the normalized trend in the therapeutic group, and better efficacy was observed in the M-Lu group. 

Neuronal injury is the major cause of long-term disability in cerebral ischemia stroke patients. Therefore, the neuroprotective effects of Lu or M-Lu were evaluated using Nissl stain and the TUNEL of the brain tissues. As shown in Figure 8B (Nissl stain) and Figure 9 (TUNEL stain), neuronal injury and apoptosis occurred at the ischemic penumbra in the experimental groups. The number of the Nissl corpuscles in M-Lu group was the lowest, and there were more TUNEL-positive neurons than in the free Lu groups, which might be due to the higher concentration of M-Lu in the brain. These results suggest that M-Lu could repair the damage to the brain tissue and inhibit nerve apoptosis.

### 3.10. In Vivo Anti-Inflammation and Anti-Oxidative Stress Response 

Inflammation and oxidative stress are the two major mechanisms in IR injury. According to the results of in vitro anti-inflammation and anti-oxidative stress testing, Lu affected the MAPK and NF-κB pathway to mitigate cellular injury. Therefore, to further verify the in vivo mechanism of the protective effect, the inflammatory mediator cytokine and oxidative stress indexes in brain tissues were detected. As shown in Figure 10A, the red fluorescence density in each group indicated that the expression of TNF-α, IL-6, and IL-1β were all down-regulated after administration. For oxidative stress (Figure 10B), the activity of SOD and GSH-px were higher and the expression of MDA was lower in free Lu and M-Lu group than in the IR group, which suggests that Lu could alleviate the oxidative stress response in ischemic penumbra. In summary, the anti-inflammatory and anti-oxidation effects of M-Lu were superior to those of free Lu. Briefly, M-Lu could repair IR injury through the in vivo inhibition of the inflammation and oxidative stress response injury.

## 4. Conclusions

In summary, we analyzed the composition of Ruyi Zhenbao pill in a classical anti-stroke Tibetan prescription, and Lu was recognized as one of the major active components. We improved the disadvantages of Lu by loading Lu into MPEG-PLGA to form micelles. The characterization of micelles showed its solution dispersibility and sustained drug release behavior. Based on the drug delivery experimental results of Lu-micelle, the micelle enhanced the cell uptake and the drug concentration at the injury site. This treatment for anti-IR injury suggests that M-Lu maintained better protective performance than free Lu. The inhibition of inflammation and oxidative stress and the induction of the differentiation of stem cells were the important mechanisms through which Lu repairs IR injury. In conclusion, M-Lu could be a candidate for IR injury treatment.

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
