# Peer review of "Pharmacodynamic Effect of Luteolin Micelles on Alleviating Cerebral Ischemia Reperfusion Injury"

_pharmaceutics, 2018, doi:10.3390/pharmaceutics10040248_

Round 1
Reviewer 1 Report
In this manuscript Tan et. al. describe the development of Luteolin micelles (M-Lu) for applications in the management of cerebral ischemia reperfusion injury (IR). The authors describe the synthesis, characterization as well as in vitro and in vivo testing of M-Lu for IR management. The manuscript is somewhat difficult to read and understand in its current form. Following are some suggestions to improve the manuscript:
1. There is a grammatical error in the title of the manuscript. Please fix.
2. The manuscript needs to be edited by a native English speaker for language and grammar to improve readability.
3. Some figures are difficult to read because of their extremely small size e.g. Fig 2A, 2C, 3B, 7B, 7C, 10 B. etc. The authors should carefully review all figures for font sizes (especially the axes labels) and legibility.
4. In Figure 2, its unclear what the authors mean by the release of "free Lu". Is this release of Lu from the dialysis bag? How is that relevant to the work?
Author Response
Responses to the comments:
Thank you very much. We have revised the whole manuscript following your advice. The language has been further polished. And the corrections have been used the "Track Changes" function in Microsoft Word in the revised manuscript. The followings are the responses to the reviews’ comments.
Reviewer 3. Comments and Suggestions for Authors
In this manuscript Tan et. al. describe the development of Luteolin micelles (M-Lu) for applications in the management of cerebral ischemia reperfusion injury (IR). The authors describe the synthesis, characterization as well as in vitro and in vivo testing of M-Lu for IR management. The manuscript is somewhat difficult to read and understand in its current form. Following are some suggestions to improve the manuscript:
1. There is a grammatical error in the title of the manuscript. Please fix.
------Thank you very much for your suggestion. We are very sorry for this stupid mistake and have corrected the title in revised manuscript.
2. The manuscript needs to be edited by a native English speaker for language and grammar to improve readability.
------Thanks for your advice. We have carefully corrected the grammatical and structural errors in revised manuscript, and the language has been further polished.
3. Some figures are difficult to read because of their extremely small size e.g. Fig 2A, 2C, 3B, 7B, 7C, 10 B. etc. The authors should carefully review all figures for font sizes (especially the axes labels) and legibility.
------Thank you very much. The font size and legibility in figures have been adjusted to the right size, and the high-quality images have been added in revised manuscript
4. In Figure 2, its unclear what the authors mean by the release of "free Lu". Is this release of Lu from the dialysis bag? How is that relevant to the work?
------Thank you very much. Because of the hydrophobicity of luteolin, we dissolved it in DMSO for the further research. In this study, the free Lu was short for the DMSO solution of luteolin, which was used as the reference preparation for M-Lu. The description has been added in revised manuscript.
Thank you again for your valuable comments and suggestions.

Reviewer 2 Report
Tan et al., demonstrated that the Lu micelles (M-Lu) enhance the therapeutic effect in IR injury restoration. The author further investigated the M-Lu could also induce stem cells to differentiate into neuron-like cells to promote the repair and regeneration of neuron. This study is very well designed and executed with multiple experiments evidence.
Luteolin-loaded micelles stability is a critical factor for further applications, and I would recommend testing in various pH, Buffer and biological media. Here some investigator used set protocol for particles stability testing. Biomaterials; 185, 2018, 174-193.
Minor comments
"The drug release behaviors in vitro" should be in vitro drug release study.
"Cellular uptake" should be cell uptake study.
Figure 9 and 10; panel A; must be replaced with high-quality images.
Author Response
Responses to the comments:
Thank you very much. We have revised the whole manuscript following your advice. The language has been further polished. And the corrections have been used the "Track Changes" function in Microsoft Word in the revised manuscript. The followings are the responses to the reviews’ comments.
Reviewer 2. Comments and Suggestions for Authors
Tan et al., demonstrated that the Lu micelles (M-Lu) enhance the therapeutic effect in IR injury restoration. The author further investigated the M-Lu could also induce stem cells to differentiate into neuron-like cells to promote the repair and regeneration of neuron. This study is very well designed and executed with multiple experiments evidence.
Luteolin-loaded micelles stability is a critical factor for further applications, and I would recommend testing in various pH, Buffer and biological media. Here some investigator used set protocol for particles stability testing. Biomaterials; 185, 2018, 174-193.
------Thanks very much for your suggestion. The stability of luteolin-loaded micelles was tested in normal saline, PBS (pH 7.4) and DMEM medium, respectively. The micelles could keep stable and no significant difference in particle size among the different media. These results and discussion have been added in revised manuscript.
Minor comments
"The drug release behaviors in vitro" should be in vitro drug release study.
------Thank you very much. “The drug release behaviors in vitro” has been corrected to “drug release behaviors in vitro” in revised manuscript.
"Cellular uptake" should be cell uptake study.
------Thanks for your advice. “Cellular uptake” has been corrected to “cell uptake study” in in revised manuscript.
Figure 9 and 10; panel A; must be replaced with high-quality images
------Thank you very much. The high-quality images have been added in revised manuscript
Thank you again for your comments and suggestions.

Reviewer 3 Report
Comments:
This manuscript, ID pharmaceutics-388302, titled “Pharmacodynamic Effect of Luteolin Micelles on Alleviates Cerebral Ischemia Reperfusion Injury" for the Pharmaceutics Journal. It presents a study on the use of a micelle formulation for delivering luteolin for treating cerebral ischemia reperfusion injury. The formulation efficacy was assessed in vitro and in vivo, using an occlusion of middle cerebral artery rat model for the latter. In this study the research strategy was appropriate, with the methods clearly described, the results clearly presented and discussed, properly supporting the conclusions reached. This work should be fit for publication after the following revisions:
1. The overall quality of language needs to be improved, including revision of grammatical errors throughout the document. I suggest reading the manuscript again and making the necessary corrections. These errors MUST BE CORRECTED.
2. Revise the title with “Pharmacodynamic Effect of Luteolin Micelles on Alleviating Cerebral Ischemia Reperfusion Injury."
3. I believe that “traditional ethnic formulation” is a more descriptive term than “ethnic formulation.”
4. Figures 2–5, 7-10: The meaning of the values (e.g., mean), errors and number of samples (n) need to be described.
5. Table 1: The meaning of the values (e.g., mean), errors and number of samples (n) need to be described.
Author Response
responses to the comments:
Thank you very much. We have revised the whole manuscript following your advice. The language has been further polished. And the corrections have been used the "Track Changes" function in Microsoft Word in the revised manuscript. The followings are the responses to the reviews’ comments.
Reviewer 1. Comments and Suggestions for Authors
Comments:
This manuscript, ID pharmaceutics-388302, titled “Pharmacodynamic Effect of Luteolin Micelles on Alleviates Cerebral Ischemia Reperfusion Injury" for the Pharmaceutics Journal. It presents a study on the use of a micelle formulation for delivering luteolin for treating cerebral ischemia reperfusion injury. The formulation efficacy was assessed in vitro and in vivo, using an occlusion of middle cerebral artery rat model for the latter. In this study the research strategy was appropriate, with the methods clearly described, the results clearly presented and discussed, properly supporting the conclusions reached. This work should be fit for publication after the following revisions:
1. The overall quality of language needs to be improved, including revision of grammatical errors throughout the document. I suggest reading the manuscript again and making the necessary corrections. These errors must be corrected.
------Thanks very much for your advice. we have carefully revised all the typo-grammatical errors in this manuscript, and the language has been further polished.
2. Revise the title with “Pharmacodynamic Effect of Luteolin Micelles on Alleviating Cerebral Ischemia Reperfusion Injury."
------Thank you very much, we have corrected the title of this paper in revised manuscript.
3. I believe that v is a more descriptive term than “ethnic formulation.”
-------Thanks for your advice, “ethnic formulation” has been modified into “traditional ethnic formulation” in revised manuscript.
4. Figures 2–5, 7-10: The meaning of the values (e.g., mean), errors and number of samples (n) need to be described.
------Thank you very much. The detail descriptions for the figures have been supplemented in revised manuscript.
5. Table 1: The meaning of the values (e.g., mean), errors and number of samples (n) need to be described.
------Thank you very much. The detail descriptions for Table 1 have been supplemented in revised manuscript.
Thank you again for your valuable comments and suggestions.
